# Functional Two-Dimensional Materials for Bioelectronic Neural Interfacing

**DOI:** 10.3390/jfb14010035

**Published:** 2023-01-07

**Authors:** Mohammad Karbalaei Akbari, Nasrin Siraj Lopa, Marina Shahriari, Aliasghar Najafzadehkhoee, Dušan Galusek, Serge Zhuiykov

**Affiliations:** 1Department of Solid-Sate Sciences, Faculty of Science, Ghent University, Krijgslaan 281/S1, B-9000 Ghent, Belgium; 2Faculty of Bioscience Engineering, Ghent University Global Campus, 119-5 Songdomunhwa-ro, Yeonsu-gu, Incheon 21985, Republic of Korea; 3Gachon University Gil Medical Center, Incheon 21565, Republic of Korea; 4Joint Glass Centre of the IIC SAS, TnUAD and FChPT STU, 911 50 Trencin, Slovakia; 5Centre for Functional and Surface Functionalized Glass, Alexander Dubček University of Trenčín Študentská, 911 50 Trencin, Slovakia

**Keywords:** two-dimensional materials, neural interfacing, neural electrodes, bioelectronic systems

## Abstract

Realizing the neurological information processing by analyzing the complex data transferring behavior of populations and individual neurons is one of the fast-growing fields of neuroscience and bioelectronic technologies. This field is anticipated to cover a wide range of advanced applications, including neural dynamic monitoring, understanding the neurological disorders, human brain–machine communications and even ambitious mind-controlled prosthetic implant systems. To fulfill the requirements of high spatial and temporal resolution recording of neural activities, electrical, optical and biosensing technologies are combined to develop multifunctional bioelectronic and neuro-signal probes. Advanced two-dimensional (2D) layered materials such as graphene, graphene oxide, transition metal dichalcogenides and MXenes with their atomic-layer thickness and multifunctional capabilities show bio-stimulation and multiple sensing properties. These characteristics are beneficial factors for development of ultrathin-film electrodes for flexible neural interfacing with minimum invasive chronic interfaces to the brain cells and cortex. The combination of incredible properties of 2D nanostructure places them in a unique position, as the main materials of choice, for multifunctional reception of neural activities. The current review highlights the recent achievements in 2D-based bioelectronic systems for monitoring of biophysiological indicators and biosignals at neural interfaces.

## 1. Introduction

The rapid advancements in neuroscience and bioinspired neurotechnologies require the accurate monitoring, reception and decoding of neurological potential signals of the nervous system [1]. In this technology, the main function is the precise recording of neuron signals through biocompatible sensors or so-called neural interfaces [2,3]. The neural interfaces execute the key functions for unlocking the performance and the governing mechanisms of neural circuits [4]. Originated from Hans Berger’s 1929 idea, the interconnection of brain signals into electroencephalogram (EEG) devices opened up enormous opportunities for novel and unique technologies similar to human–robot interfaces and brain-controlled autonomous vehicles [5,6,7]. Therefore, neural interface devices have been developed to dynamically interact with the human nervous system (Figure 1a) for a wide array of applications ranging from deep brain stimulation to mind control. The neural interfaces are also employed extensively in clinical therapy for treatment and monitoring of neurological disorders such as neurobiological depression [8], autism [9], epilepsy [10], sensorimotor dysfunctionality [11,12], and Alzheimer's [13] and Parkinson disease [14].

The advent of the first generation of neural interfaces was considered as a key development toward the realization of electrical properties of cell and tissues that enabled in vitro neural signal acquisition. The voltage clamp technology, developed in 1976 by Erwin Neher, enabled us to record single ion channel signals [2]. In 1981, the advent of gigaseal clamps facilitated the reception and recording of ultra-low neural signals. The current patch clamp technology is still able to simultaneously and effectively monitor and record the signals of single ion channels [2]. Despite the capabilities of the first generation of neural interfaces, the sample preparation and measurements were still time-consuming and complicated. The multi-channel neural interfaces are the second generation of these devices, enabling in vivo measurements of neural activities. In the second generation of neural interfaces, the in vivo electrodes are connected into the data acquisition setup via cables. The second generation of neural interfaces let us simultaneously stimulate the neural system and monitor the generated signals [2]. The wireless connection enabled the development of a novel generation of neural interfaces or so-called third-generation devices. This type of wearable/implantable neural interface can be integrated into a signal acquisition system via a wireless communication unit. This facilitates the free movement of the subjects. In this system, the recording electrodes are separated from the sensing modules. In the latest generation of neural interfaces, or the so-called fourth generation, the electronic sensors are integrated with low-power amplifiers and signal processing units on a single platform to develop an ultra-compact low-power electronic neural interface system.

Neural signal recording can be divided into two main categories, i.e., intercellular and extracellular recordings. The intracellular technique records the electric signals generated across the cell membranes; therefore, the electrodes are placed inside and outside of the membranes. In a different configuration, extracellular recording is enabled through the employment of recording electrodes in the extracellular fluid for the reception of signals generated by nearby neurons. There are several main differences between intracellular and extracellular recording. The intracellular neural signals are more localized and feature more powerful amplitudes, while the extracellular signals are weaker and originate from a large area of the brain. The extracellular recording of neural signals can be divided into two main categories of invasive and non-invasive methods based on the location of electrodes. In the non-invasive neural interfacing, the signals are collected via electrodes assembled on the top surface of the scalp (Figure 1b) to record EEG signals of neural activities. The non-invasive recording of neural signals imposes a low level of health risk to users [15,16]. For instance, external flexible EEG neural interfaces enable the low-risking record of neural signals through wearable sensors for the monitoring of brain health activities and early diagnosis of neurodegenerative diseases [17,18,19,20,21,22,23,24]. The non-invasive neural interfacing by biocompatible, wearable electrodes is employed along with other clinical brain signal monitoring and medical monitoring systems such as magnetic resonance imaging (MRI), computed tomography (CT) scans and intracranial electroencephalography (iEEG) [25,26].

The invasive neural interfaces can be categorized into penetrating and non-penetrating sensors (Figure 1c–e) [27,28,29]. The most famous invasive brain neural interface electrodes are electrocorticography (ECoG) sensors where the electrodes are inserted under the skull and on top of the cerebral cortex (Figure 1f). The epidural ECoG electrodes are assembled on the dura mater, while the subdural ECoG electrodes are placed on the arachnoid [2]. Micro-electrocorticography (μECoG) is another technology that offers higher levels of spatial and temporal resolution compared with those of conventional ECoGs. In the μECoG, the miniaturized electrodes can be assembled even on the soft substrates and, therefore, enable more flexibility for neural recording. The size of a conventional ECoG electrode covers an area of 1 cm, while a flexible μECoG can enhance the spatial resolution within the sub-millimeter range [2]. The flexibility of μECoG enabled us to minimize the distance between the sensing electrodes originated from the geometrical adaptability of these novel sensors. Regarding the continuous development of micro and nanofabrication technologies, it is expected that the spatial resolution can be enhanced considerably in the novel ECoG transistors. To measure the local field potential (LFP) of neural activities, it is necessary to investigate the inside of brain tissue via intercortical electrodes (Figure 1c,d) [30,31,32]. The conventional biocompatible metal-based microwire electrodes are extensively employed for intercortical neural interfacing where the advanced techniques enable the monitoring of input and output interactions of one million neurons with a tightly bounded microwire electrodes [2]. However, the crosstalk between channels and signal interference are the main challenge of this typical invasive neural interface system. Another replacement is the complementary metal oxide semiconductor (CMOS) based microelectronic device developed on silicon-based needles. Despite the precise performance of these neural interfaces, the tissue damage caused by the rigid probe may enhance the inflammation and, therefore, adversely degrade the signal recording capabilities of neural interface sensors. The replacement of rigid substrates with more flexible polymer-based probes is another approach to minimize the tissue damage. While the flexible electrodes are more friendly and biocompatible with human tissue, the successful implantation of invasive flexible electrodes is more complicated and requires the employment of robotic systems for the insertion procedure [2]. The minimization of the footprints of the electrodes is the main strategy to reduce the tissue response. Consequently, micrometer-sized carbon-based fiber electrodes have been developed for intercortical neural interfacing to reduce the footprints to the minimum level.

From a technical point of view, an efficient and reliable neural interfacing process can be achieved through the development of biocompatible sensors with high signal-to noise-ratio (SNR) characteristics and low and stable impedance values [33]. Figure 1f depicts different types of invasive and non-invasive neural interfaces developed for monitoring EEG and EcOG signals of the brain. In this area, various types of invasive and non-invasive neural interfaces must overcome different technical challenges to fulfill the requirements for reliable neural interfacing performance. The main challenges for application of implanted invasive sensors originate from corrosion, inflammation, toxicity and biocompatibility issues. The early types of conventional invasive neural sensors are mostly based on wet systems where metal electrodes record the informative neural signals. Ag/AgCl electrodes are among the most common conventional low-impedance wet electrodes for invasive neural interfacing [34]. However, the invasive sensors based on Ag/AgCl electrodes suffer from several fundamental deficiencies, including biocompatibility issues, time-consuming and complicated implantation procedures, and unstable performance in non-clinical conditions in moveable organs [31,32]. Soft ionic hydrogel-based electrodes use gel-paste compounds in non-invasive neural sensors; therefore, the amount of liquid components is decreased. These non-invasive electrodes have performance similar to that of conventional wet-gel electrodes in terms of short-circuit noise, electrical impedance, and skin–electrode contact impedance. Consequently, their peripteries are significantly better than those of water-based electrodes. Nevertheless, the wet nature of hydrogel sensors raises some serious concerns about their reliable long-term biosafety performance [35]. Therefore, considerable attention has been devoted to the development of dry sensors for neural interfacing applications [36,37,38]. The current dry neural sensors are mostly based on multi-conductive mesh-like metal sensors tailored on foam holders. However, both the dry and wet sensors cannot sufficiently perform under harsh conditions. The main factors that limit the functionalities of current neural sensors include insufficient SNR values, fidelity, long-term stability issues and high impedance values. Despite considerable development, biocompatibility is still another main challenge in neural interfacing. Specifically, the natural mismatches between conventional invasive neural interfaces and brain tissues cause severe inflammatory responses [39]. Another main limitation is related to the geometrical features of electrodes for reception of potential signals from individual neurons. Micro-sized invasive neural sensors have difficulties in performance due to increased electrochemical impedance at heterointerfaces. It results in a corresponding intensification of thermal noises, signal interference and losses. Consequently, these problems degrade the stable, long-term performance of neural interface sensors [40]. Thus, ground-breaking achievements in the field of brain–machine interfacing are hindered due to the technological limitations [41,42]. Overcoming the limitations of high impedance and electrode footprints requires efforts to enhance the sensors’ surface area, and consequently, increase the possibility of interaction and reception of ionic species at the neural interfaces. From a functional point of view, the main strategy is the enlargement of the active surfaces at neural interfaces via employment of nanomaterials and nanopatterns and increasing the surface roughness and porosity content of nanostructures [43,44,45,46,47,48].

Studies on multifunctional biosensing probes have fueled a continuous need for new materials. These materials are expected to combine multiple sensing and stimulation functionalities and simultaneously create chronic biocompatible interfaces with neural cells and brain tissue. In this regard, 2D nanomaterials are among the main advanced atomically thin nanostructures that hold great promise for the development of highly active and miniaturized sensors [49,50,51,52]. Due to their high surface area, easy functionalization and exceptional electronic characteristics, 2D nanostructures are among the main options for the development of novel neural interface sensors [3,53,54,55,56,57]. The atomically thin structures of 2D materials make them intrinsically flexible structures, allowing for applications in soft adaptable bioelectronic systems. Consequently, 2D nanostructures are the main options for the establishment of mechanically firm and reliable, yet flexible and electronically adaptive structures for sensing of indicator signals in biological tissues. In contrast to the silicon-based technologies, 2D materials can bring essential advancements to flexible neural interfacing technology via their distinguished electronic characteristics, transparency and biocompatibility. These flexible structures can be assembled either as invasive or non-invasive sensors for monitoring of neural activities. Wearable brain–machine interfacing technologies extensively employ imperceptible micro-needles and flexible circuits to improve neural signal recording. The desired wearable neural interfaces are expected to be assembled externally and provide the minimum invasive component. Two-dimensional materials can be successfully employed as the sensing or complimentary components of these wearable neural interfaces. Nevertheless, there are still functional and scientific challenges to the facile application of 2D materials as the main components of neural interfaces. Despite the promising functionalities of 2D materials, biocompatibility issues and long-term performance are among the main challenges that still need to be overcome. The current review focuses on the latest advancements in the development of neural interfaces based on 2D nanomaterials, including graphene, graphene oxide (GO), reduced graphene oxide (rGO), transition metal dichalcogenides (TMDCs), and MAX phase (MXenes) 2D materials. Therefore, this paper provides an overview on the functionalities of such 2D nanomaterials for their application in neural interface systems. Figure 1g schematically depicts different types of 2D nanomaterials employed as the main sensing components of neural interfaces and bioelectronic sensing systems. It is worth mentioning that these 2D based sensors were also employed simultaneously for monitoring of other biological and physical signals of the human body, including electrocardiography (ECG or heartbeat), electromyography (EMG, or muscle contraction), and electrooculography (EOG or eye movement) signals. However, this review specifically covers the latest achievements in the development of neural interface technologies where 2D materials are employed for the monitoring of brain neural activities (EEG and EcOG).

## 2. Carbon Based 2D Materials

### 2.1. Garphene

Graphene, the most famous carbon-based 2D structure, is recognized as a functional material for neural interfacing applications [53]. Its excellent conductivity, mechanical stability, transparency and facile functionalization place graphene in a unique position for applications in neurotechnologies, biosensing and multi-modal neural interfacing [58,59]. Graphene-based 2D materials have been employed either passively or actively as the components of microelectrodes or transistors in neural interfaces. The graphene neural interfaces record either chemical signals, such as the concentrations of neurotransmitters, or measure electrical indicators, such as local field potentials. Regarding the design and working mechanisms of graphene-based neural interfaces, the next sections overview the latest discussions on graphene applications in neural interface sensors.

High-temporal-resolution recording of neuron cell activities is one of the main targets of electrophysiology, where optical and cellular imaging techniques are accompanied by electrophysiological signal recordings to complement the performance of each individual technique and then provide a clear image of the neurological behavior [60,61,62]. However, conventional metal microelectrodes cannot be employed in cellular imaging techniques due their shadow and blockage effects [63,64]. Furthermore, light-induced effects on the metal components adversely interfere in the potential signal recording process. These limitations can be overcome by the development of transparent 2D graphene microelectrodes, where the optical transparency of the graphene-based neural interface enables cellular imaging and electrophysiological recording [65]. Different synthesis methods for 2D graphene provide various types of 2D nanostructures. Chemical vapor deposition (CVD) graphene [66], few-layer exfoliated graphene, and chemically synthesized graphene all have different physical and chemical characteristics. The availability of various 2D graphene nanostructures with distinct properties and doping capabilities have opened up outstanding opportunities for the development of neural interfaces. Graphene field-effect transistors (GFETs) [67] are another electronic system developed for neural interfacing applications. GFETs have interesting applications in bioelectronic systems. The next section will review various types of graphene-based neural interfaces.

#### 2.1.1. Graphene Microelectrodes

Fluorescence microscopy can be combined with invasive graphene-based microelectrodes to visually depict the precise active location of neural signals [63]. Calcium imaging with fluorescence response cannot resolve the high-frequency spikes of large populations of neural cells, while the presence of graphene-based neural interface electrodes enabled the detection of high-frequency neural reactions [63]. A study investigated the simultaneous electrophysiological recording of neural responses via invasive transparent graphene-based neural interface electrodes. The results were later combined with optical fluorescence imaging. This approach incorporated the advantages of temporal and spatial resolution of both electrophysiological recording and imaging techniques (Figure 2a–c). Such an approach ensures detection of the internal and ictal activities, and also guarantees the successful monitoring of ultra-fast neural spikes (5 ms) and population discharges, which are not possible to track by using merely the multicellular calcium imaging technique [65]. This developed graphene-based microelectrode system also demonstrated a six-fold improvement in the SNR value and 100-fold reduction of electrical interference noise [65]. Similar results were also reported by another study where electrical recording synchronized with wide-field calcium imaging was combined for investigation of the local neural activities of cortical modules [65]. A comparison of interictal-like spiking activities recorded by Au and graphene electrodes confirmed that the noise level in the doped graphene electrodes was almost six times lower than that of Au electrodes, while the impedance levels remained comparable at a signal frequency of 1 kHz (Figure 2f). Subsequent comparisons showed the suppression of electrical interference noise in doped graphene electrodes. This confirmed that graphene has a clear advantage for the study of brain activities at low local-field potential ranges (1–100 Hz), which are a property of information pulses and synaptic potential signals at low frequencies (Figure 2g). Another study employed four-layer chemical vapor-deposited (CVD) graphene films on transparent invasive electrode arrays to successfully monitor neural activities in a mouse neural system expressing channelrhodopsin-2 under blue light stimulation [56]. Transparency and thickness were two interrelated factors that already affected the impedance values of optically stimulated neural interfaces in this study [68]. By the careful design of transparent graphene electrodes (number of layers), it was possible to tackle the challenges stemming from the light-induced problems to achieve local-field potential recording of neural activities via optogenetics stimulation [69]. For example, the graphene monolayer is transparent, but its impedance is relatively high compared with that of other types of conventional porous structures and metal faradic interfaces. The proposed mechanism for the impedance reduction was an alteration of graphene surface characteristics by nitric acid (up to almost 50%) [67,70]. Furthermore, there are always trade-offs among the electrode footprint, SNR, and interference impedances at the neural interfaces. An increase in contacting interfaces can facilitate the capture of a higher number of ionic species transferred at the heterointerface between the graphene electrode and extracellular spaces. Despite the numerous desirable characteristics of graphene monolayer for neural interfacing, the intrinsic capacity of graphene monolayer for charge injection is still not sufficient for the efficient electrical stimulation of neurons. It was suggested that porous 3D structures of graphene nanosheet electrodes can efficiently demonstrate high capacity for charge injection at the neural interfaces and at the same time present reasonably low heterointerface impedance [45]. Favorably, the fabrication of 3D porous graphene-based electrodes with polymer coatings considerably enhanced the mechanical stability and reduced the impedance of neural interfaces [71].

As an example, porous graphene nanostructures were grown on polyamide substrate by laser pyrolysis (Figure 3a–c) [45]. This approach enabled the fabrication of highly adhesive 3D graphene nanosheets with considerable mechanical stability of electrodes for neural interfacing applications (over one million cyclic operations) [45]. A high charge capacitance value of 3.1 mC cm^−2^ was recorded for a 3D graphene-based electrode that was employed for in vivo stimulation of motor cortex arrays of mouse leg [45]. In this setup, porous graphene electrodes containing 16-electrode arrays of graphene whiskers were employed by a pair of needle electrodes to stimulate the motor cortex of mouse brain with electrical pulses (Figure 3d). These electrical signals evoked transient knee and ankle flexion in contralateral mouse legs. The corresponding electrical current response of the system was measured and recorded (Figure 3e). A direct relation was found between the amplitude of the stimulus current and the movement response of knee and ankle (Figure 3f), confirming the capability of the developed penetrating porous graphene electrode arrays as minimally invasive neural interfaces for the brain–computer interface and neural prosthesis applications [45]. In particular, the distinctive capacitive characteristics of fabricated 3D graphene arrays tangibly eliminated the corrosion-related risks that are recognized as critical challenges in faradic electrodes.

#### 2.1.2. Graphene Field-Effect Transistors (GFETs)

In parallel with microelectrodes, GFETs are being developed for a wide range of biosensing applications, where the transduction of biological signals is amplified locally due to the strong field effects induced [30,72,73,74]. The main advantage of GFET configurations compared to graphene microelectrodes stems from the impact of intrinsic amplification on the SNR factor of neural interfaces. A GFET works on the basis of the interaction of 2D graphene gate material with the electrolyte solution. A study was conducted on the visual part of brain cortex where GFET neural interfaces were employed for measurements. In this study, SNR values of 62 for pre-epileptic activities and 9.8 for spontaneous oscillation were achieved [75]. Similar values for Pt electrodes were, respectively, 53 and 8.33 for in vivo recording [75]. In another recent study, a monolayer graphene gate was developed by chemical vapor deposition to be in contact with brain tissue where the gate bias voltage was applied via a reference electrode [73]. The presence and concentration of charge carriers at the neural interfaces on brain tissue can alter the conductivity of the graphene gate upon the application of gate voltages at the graphene-electrolyte heterointerfaces. Therefore, planar arrays of 2D graphene were developed on the transparent substrate [73]. These multi-neural probes were assembled on the rat’s brain tissue for mapping targets. In this setup, the GFET arrays could continuously provide mapping images from the epicortical brain activities for 24 h by monitoring patterns of responses to infra-slow fluctuations (ISFs) [73,74]. Infra-slow oscillation of EEG signals refers to specific brain signals with frequencies in the range of 0.01 and 0.1 Hz. Figure 4 depicts a typical sensing system consisting of a flexible multichannel GFET arrays tailored to a wireless system for the facile monitoring of brain activities of moving rats within a wide range of frequencies. The neural interfaces were composed of non-invasive graphene array electrodes positioned on the cortex of a rat, where an Ag/AgCl reference electrode and two individual Pt-Ir electrodes were accompanied by GFET arrays (Figure 4b). A schematic graph of a single GFET is presented in Figure 4c. Informative signals extracted from coupled channels from the neural interface were accompanied by corresponding mapping at ultra-low frequencies (~1 MHz). The results confirmed the capability and maturity of this Internet of Things (IoT) technology for online, long-term and consistent monitoring of brain activities (Figure 4d,e) through in vitro and in vivo evaluations of neural signals [74].

### 2.2. Other Types of Graphene-Based Materials

The availability of various types of graphene-based materials, including single-layer CVD films, multilayered exfoliated films, and chemically and mechanically synthesized films, has enabled the development of various types of graphene-based neural interfaces. Despite the fact that a large number of bioresearch activities have been devoted to graphene, the high hydrophobicity of pristine graphene has limited its applications. Functionalized graphene, graphene oxide (GO) and reduced graphene oxide (rGO) 2D structures can be valuable alternatives to graphene in neural interfaces. Two-dimensional GO is an atomically thin structure where oxidation enables the incorporation of oxygen functional groups into the surface of the graphene, thus decreasing the graphene aggregation in aqueous solutions as well as reducing its electrical conductivity. Recently, ultra-large graphene oxide flakes were deposited on binder-free microfibers to replace conventional filament-shaped metallic invasive neural electrodes [64,65]. In these GO-based microfiber sensors, the specific contact impedance was improved to a lower value (3.9 MΩ μm^−2^ at the 1 kHz frequency), while it showed considerably better charge storage capacity (361 ± 45 mC cm^−2^) compared with that of the conventional metallic-based filament microelectrodes used in neural interfacing systems. Similar research studies also included graphene as the main sensing component in microfibers [76,77].

Despite its greater functionalization capabilities, GO has highly restricted electron mobility on its surface. GO is technically an insulator; therefore, reduction reactions are carried out to partially remove oxygen and then alter or recover the electrical conductivity of 2D GO films. Consequently, reduced graphene oxide (rGO) with higher conductivity is extensively employed for biosensing and especially non-invasive neural interfacing applications. Typically, 2D rGO nanostructures are developed or grown during the fabrication process for neural interfaces at the heterointerfaces between the sensing components, which in turn facilitate swift charge transfer and low contact impedance at the neural interfaces. Non-invasive graphene-based polymer wearable sensors are well-developed for the characterization and precision sensing of biosignals with high levels of accuracy [20,21,22]. One of the main and widely available polymer substrates for fabrication of non-invasive thin-film wearable electronic neural interfaces is polydimethylsiloxane (PDMS). Various conductive layers can be developed on the PDMS layers by electrospinning, sputtering and spray coating to fabricate the skin-mounting side of the neural interfaces. There are several reports of successful performance of graphene-based wearable and flexible neural interfaces, where GO and rGO were employed as the main components of the sensors. In one study, neural interfaces were composed of patterned rGO films developed from chemically treated GO nanosheets. This setup showed considerable low contact impedance of 500 kΩ at a signal frequency of 50 Hz [56]. In another example, rGO-based neural interfaces were developed on PDMS substrates through laser-assisted GO reduction. These flexible non-invasive sensors demonstrated ultra-low skin impedance of ~60 kΩ at 10 Hz [78]. Another rGO/nylon epidermal sensor was fabricated by the hydrothermal synthesis of GO, followed by its thermal reduction to rGO films. Such conductive sensors with low on-skin impedance of 15 kΩ at 100 Hz were used for non-invasive monitoring of ECG and EMG signals [79]. Another type of non-invasive neural interface are E-skins, which are soft and flexible epidermal electronic devices with multifunctional applications. Graphene and its family were extensively employed for the fabrication of tattoo sensors for the reception and sensing of vital neural and physical signs [80,81,82]. Specifically, e-tattoos based on the 2D graphene family with nanomesh structures [83] and CVD graphene [84] provided considerably low resistance (~24 Ω/sq) for precise reception and long-term monitoring of brain activities (EEG) and sleep monitoring. In this setup (Figure 5a–d), optimized dry electrodes were fabricated by CVD of graphene layers with thickness of 100 nm on PEDOT:PSS electrodes (PTG). These electrodes exhibited high levels of conductivity attributable to their graphene layers (4142 S/cm), high optical transparency (Figure 5c) and mechano-electrical stability for e-skin bioelectronic applications [85]. Optical transparency enabled incorporation of the outcomes of laser speckle contrast imaging into EEG and EMG measurements. This process was highly beneficial for the deep understanding of pathological process mechanisms (Figure 5d). The developed PTG sensors successfully recorded EEG signals in the sleeping, exercising and relaxing states (Figure 5e).

Due to the facile dispersion of GO and rGO, these structures can be feasibly used for the fabrication of fiber neural interfaces. In such development, the fibers (nylon, cotton, polyesters) are dip-coated in GO or rGO dispersions, followed next by thermal treatment. These microfiber neural interfaces showed considerably improved conductivity and low impedance for wearable neural interfacing [85,86,87,88,89]. Monolayer and porous graphene were also employed for free-standing neural interfaces. In one approach, liquid-crystal GO and rGO were used to make microfiber sensors with a high level of charge injection capacity (46 mC cm^−2^) [89]. These free-standing electrodes based on reduced liquid-crystal GO and porous graphene electrodes successfully stimulated retinal ganglion cells. The incredibly low impedance, high charge injection, storage capacity and high fidelity of these sensors for neural stimulation originated from their large effective surface areas. Improvements in the conductivity of rGO have been extensively investigated to enhance the functionalities of these 2D structures. Chemical treatment is one of the main approaches. As an example, a low-temperature chemical method using hydroiodic acid (HI) and L-ascorbic acid (L-AA) can effectively enhance the conductivity of rGO. While HI and L-AA are known as effective reductants, their combination resulted in improved electrical conductivity of up to 1115 S/m [90]. Another study employed graphene oxide–gold oxide (rGO/Au_2_O_3_) nanocomposite electrodes as implantable neural interfaces for applications in electrophysiology and neurochemistry. The rGO/Au_2_O_3_-modified electrodes displayed significantly improved activities [91]. Current studies are devoted to improving the functionalities of graphene-based 2D neural interfaces to attain high spatiotemporal resolution signal recording. The majority of existing sensors work on the basis of electrochemical sensing of biosignals. Low heterointerface impedance accompanied by high SNR characteristics are the main requirements for graphene-based neural interfaces. Table 1 provides a brief overview on the neural interfaces fabricated by graphene, GO and rGO 2D materials.

## 3. Two-Dimensional Materials beyond Graphene

Two-dimensional materials beyond graphene represent atomically thin nanostructures with remarkably novel semiconducting, insulating and metallic characteristics that put them in a distinguished position in comparison with their carbon-based counterparts. The family of 2D materials beyond graphene covers various type of nanostructures, transition-metal dichalcogenides (TMDCs) and MAX phase 2D materials (MXenes). These 2D structures recently found their way into neural interfacing technologies [96]. The following sections overview recent advancements in neural interface technologies based on 2D materials beyond graphene.

### 3.1. Transition-Metal Dichalcogenides

TMDCs are the well-known and perhaps the most eminent 2D materials beyond graphene [97]. TMDCs are recognized by their generic MX_2_ chemical formula, where M is one of the transition metals (Mo, W, V, Pt, etc.), and X is one of the chalcogen elements (S, Se, Te). The electronic characteristics of 2D TMDCs are of particular interest, since they differ dramatically from those of their bulk and multi-layered structures and even their single-layered 2D structures [97,98,99]. Electronic structural dependency aside, the properties of 2D TMDCs are also related to their chemical composition. For example, platinum (Pt) is well-known for its corrosion resistance, catalytic characteristics and biocompatibility [100,101,102]. The incorporation of this noble metal with chalcogenide elements enabled the synthesis of 2D TMDC nanostructures with distinct biocompatibility characteristics. Furthermore, thickness-dependent semiconducting to metallic transition of 2D Pt-TMDC structures was recently reported [101,102], presenting an opportunity for the fabrication of highly biocompatible, corrosion-resistive, low-impedance and wearable neural interfaces based on 2D Pt-based TMDC nanostructures.

Recently, two types of Pt-based TMDCs, i.e., PtSe_2_ and PtTe_2_, have attracted researchers’ attention due to their electronic characteristics. The ultra-thin few-layered PtTe_2_ has shown metallic properties attributable to the strong interactions between its Te interlayer chalcogen atoms [103,104,105]. However, monolayer PtSe_2_ is generally a semiconductor material that undergoes semiconductor-to-metal transition when the number of its layers is increased [103]. Apart from their electronic properties, the synthesis and growth of 2D TMDCs on polymer-based substrate is also a challenge, and consequently, the fabrication of wearable neural interfaces can be remarkably affected by the growth process. The wafer-scaled direct growth of PtSe_2_ and PtTe_2_ ultra-thin films on polymer-based substrate at temperatures of less than 400 °C has allowed the development of flexible neural interfaces based on these 2D materials [106,107,108,109,110]. A recent study [111] developed multipurpose wearable non-invasive electronic tattoos for the detection of EEG, heartbeat, EMG, ECG, and EOG signals of the human body (Figure 6a). In these neural interface devices, PtSe_2_ and PtTe_2_ were initially grown by CVD on Si/SiO_2_ or Kapton substrate at 400 °C. The developed films on Si/SiO_2_ were afterwards transferred to 200 nm thick poly methyl methacrylate (PMMA) flexible substrate to fabricate wearable sensors (Figure 6b). The neural interface devices were composed of ~25 ± 3 nm thick multilayer 2D Pt-TMDCs developed on either Kapton or PMMA flexible substrates (Figure 6c). Au soft electrodes were further employed for the electrical connection of 2D Pt-TMDC sensors to the human skin (Figure 6c). The heterointerface between 2D Pt-TMDC neural sensors and the human skin initially experienced high impedance values owing to the presence of trapped air and microporous structures. Overall, it took about 5–10 min to make direct contact between 2D Pt-TMDCs and the human skin (Figure 6d). The normalized impedance and interface capacitance values of 2D PtTe_2_ based sensors were found to be the most distinguishing characteristics of these neural interfaces. The highest interface capacitance of 62.2 ± 8.2 μFcm^−2^ and the lowest normalized impedance of 35.4 ± 18.6 Ω·cm (Figure 6e) were recorded at 1 kHz [111]. Generally, the PtTe_2_ tattoos displayed 100 times smaller sheet resistance and about four times lower impedance values than those of 2D graphene-based neural sensors. The following measurements confirmed that neural interfaces based on 2D PtTe_2_ nanostructures displayed higher interface capacitance values compared with those of Pt, PtSe_2_ and Au based electrodes (Figure 6f). It was further realized that the SNR value (Figure 6g) of natural interfaces with 2D PtTe_2_ electrodes (84 ± 5) was higher than that of Ag/AgCl based neural interfaces (61 ± 5). On the other hand, on-skin transistors are the main wearable neural interface bioelectronic technology, enabling high-definition monitoring of the local amplification of biosignals and generated electrophysiological indicators [112]. In a recent study, a stretchable and free-standing electronic neural interface was developed based on 2D MoS_2_ TMDC films [113]. These 2D MoS_2_ based free-standing thin films with breathable capability were able to monitor electrophysiological signals. The bond-free van der Waals (vdW) forces between the 2D layers of MoS_2_ films made it possible to fabricate free-standing 2D MoS_2_ devices. The weak vdW bonding forces enabled the rotational and sliding motion of 2D layers and, therefore, facilitated the high level of flexibility of thin films. The free-standing vdW 2D films also showed a well-behaved linear relationship and displayed a gain in tensile strength of 43% with a measured Young’s modulus of 47.3 MPa [112]. The vdW interactions in biological assemblies ensured that the free-standing characteristics matched to the soft biological tissues and made them adaptable structures for the coverage of surface topographies of living organisms and especially the brain cortex. The highly conformal interfaces between free-standing 2D MoS_2_ films and underlayer films provided exceptional electronic functionalities for applications on living organisms [112]. The developed conformal electronic skin-gate 2D MoS_2_ transistors attained a considerable SNR of 49.8 dB during human motion. Skin-gate transistor neural interfaces developed on the basis of 2D MoS_2_ films demonstrated versatile performance for biosensing, biopotential amplified signals and motion analyzing/processing targets [112].

### 3.2. MXenes

MXenes are novel 2D materials representing a broad range of atomically thin nanostructures composed of early-transition metal carbides, nitride and carbonatites. These 2D materials are synthesized from the selective etching of MAX phase materials [113]. The general formula for MAX phases is M_n+1_AX_n_, where M belongs to the group of early-transition metals (such as Ti, Va, Nb, Mo, etc.), A is an element from group 12–16 of the periodic table, and X is either carbon or nitrogen, or both of them. The “*n*” in the atom indices can be 1, 2 or 3. While the bulk of MXenes represent metallic conductive characteristics, the 2D MXene films show various electronic properties, ranging from dielectric to semi-metal behavior. By implementing these tunable chemical characteristics, it is possible to orchestrate a combination of surface characteristics originated from the presence of surface-terminating functional groups on the 2D MXene films similar to hydroxyl (^−^OH), oxygen (^−^O), or fluorine (^−^F) groups. This prepares these 2D nanostructures for facile surface functionalization and modification. The hydrophilicity characteristics of 2D MXenes provide an excellent opportunity for solution-based processing techniques, and consequently, the need for functionalization treatments by super-acids and surfactants is eliminated [114,115,116,117,118,119,120]. These characteristics put 2D MXenes in a unique position compared with carbon-based 2D nanostructures. Apart from the processing stage, the hydrophilicity provides an excellent opportunity for the application of 2D MXenes in biosystems, electrochemical sensing, and drug delivery targets [114,115,116,117,118,119,120,121,122].

Among the various types of MXenes, the Ti-based MXenes are the most investigated 2D nanostructures. In particular, 2D Ti_3_C_2_ films demonstrated outstanding volumetric capacitance (1500 F/cm^3^ for Ti_3_C_2_T*_x_*) [119] and electrical conductance (10^4^ S/cm for Ti_3_C_2_T*_x_*) [121]. The volumetric capacitance of 2D Ti_3_C_2_ is by far higher than that of graphene (100 F/cm^3^) and graphene gel films (~260 F/cm^3^). A research study reported a fabrication approach for the development of high-resolution neural interfaces, where the electrode arrays were fabricated based on 2D Ti_3_C_2_. This solution-processing method was scalable and reliable for the fabrication of various types of non-invasive neural interfaces for the monitoring of EEG signals [122,123]. The neural interface in this study [123] was a microelectrode with 16 channels where the 2D Ti_3_C_2_T_x_ acted as the active sensing material. The 2D Ti_3_C_2_T_x_ films were spray-coated and patterned on parylene C substrates. The measurement of on-skin sensing capabilities of the micro sensors showed impedance values of ~29.78 ± 8.1 kΩ.cm^2^ at the frequency of 1 kHz during the identification of muscle activation. The measured SNR of this sensor was in the range of 39–16 dB [123]. Subsequent fabrication technology enabled the arrangement of more complicated neural sensors for more practical applications with improved characteristics. For example, a laser patterning approach was introduced to develop three-dimensional (3D) arrays of mini-electrode pillars based on MXene 2D nanostructures [124]. Specifically, MXene nanostructures were infused into cellulose, attached to a laser-patterned substrate, and then encapsulated in PDMS. The final electrodes were arranged in the form of circular arrays to acquire the EEG signals of alpha rhythms. On-skin measurements showed an ultra-low impedance value of 2.8 ± 0.9 kΩ at 1 kHz. These 3D sensors were able to detect brain signals even when they were assembled on the skin of the human head, representing a promising approach to the development of brain–machine interfaces [124].

Both invasive and non-invasive neural interfaces based on 2D Ti_3_C_2_ MXene nanosheets were recently developed for the high-resolution sensing of brain signals [125]. In this research, a facile and high-throughput fabrication technique was employed to combine the conventional photolithography pattern method with another ad hoc spin-coat and dry lift-off technique to fabricate precisely micro-patterned films of MXenes. In this technique, a colloidal solution containing 2D Ti_3_C_2_ nanosheets was used to fabricate the sensors on flexible parylene substrates (Figure 7a). Unlike carbon-based 2D nanostructures, the inherent hydrophilic properties of Ti_3_C_2_ nanosheets enabled the uniform distribution of 2D Ti_3_C_2_ in aqueous solution without using strong acidic or chemical surfactants. The hydrophilic characteristics were highly favorable for micro-patterning of the finest schemes of Ti_3_C_2_ films on biocompatible soft substrates. Using this method, both non-invasive and implantable multichannel Ti_3_C_2_ electrodes were fabricated to investigate the ECoG signals of the rat’s brain (Figure 7b). The non-invasive μ-ECoG electrode arrays were composed of both gold and Ti_3_C_2_ contacts. Of note, in the electrodes, 10-channel laminar arrays were developed with side-by-side arrangements of Ti_3_C_2_ and Au circular contacts of 25 μm in diameter. In the side-by-side configuration of Ti_3_C_2_ and Au circular channels, each electrode in a pair could plausibly record from the same adjacent neurons. The assembly of a developed μ-ECoG electrode enabled the monitoring of signals from cortical surfaces. Furthermore, the insertion of μ-ECoG intracortical sensors made it possible to record signals from the deeper part of the brain (Figure 7c). Both non-invasive and intracortical configurations of these μ-ECoG sensors were found to be capable of detecting physiological signals. Measurements of ECoG signals by Ti_3_C_2_ based microelectrodes demonstrated a high level of consistency among the recorded signals from different channels (Figure 7d). It was observed that the μ-ECoG electrodes recorded the brain’s signals with lower baseline noise levels, higher signal-to-noise ratios and reduced susceptibility compared with Au based electrodes. A typical example of neural activities recorded via 10-channel intracortical probe is presented in Figure 7e. Overall, the Ti_3_C_2_ electrodes achieved a four-fold reduction in interference impedance. Later studies on the biocompatibility of the Ti_3_C_2_ sensors confirmed that cultured neurons on the Ti_3_C_2_ nanostructures were alive and could adhere and grow functional neural networks. This observation confirmed the functional biocompatibility of 2D Ti_3_C_2_ MXene for high-resolution and precise neural interfacing [125]. Research works in the field of neural interfaces based on 2D MXene-based nanostructures are in their early stage. Therefore, new achievements are highly expected in the near future. Table 2 provides a brief overview of the neural interfaces fabricated by 2D materials beyond graphene.

## 4. Future Opportunities and Challenges

The further development of neural interfaces based on 2D nanostructures requires that more reliable evidence be extracted from practical applications of developed bioelectronic neural sensors. The following subsections address the main challenges and also future opportunities in the development of neural interfaces.

### 4.1. Impedance and SNR

Electrical properties such as contact impedance and SNR are still the main concerns in the development of novel flexible neural interfaces. The detection of μV amplitude neural signals requires new materials with high conductance and capacitance. The electrochemical sensing of neuro-potential signals of brain cortex is currently possible with low-contact impedance sensors. Commercial Ag/AgCl wet sensors are extensively employed for neural interfacing where the lowest non-invasive on-skin impedance of ~30 ± 5 kΩ at 100 Hz has been achieved [127,128]. Nevertheless, dry and wearable sensors are desired neural interfaces for novel technologies, including brain–computer interfacing. For this purpose, CVD graphene on conductive flexible polymers and porous graphene-based 2D structures achieved comparable impedance values during measurements on human skin. Values of ~80 kΩ and ~17 kΩ were reported, respectively, for neural interfaces fabricated from epitaxial graphene and porous graphene. To achieve even lower contact impedance at the neural interfaces, it is necessary to simultaneously develop both novel micro and nanofabrication technologies and also expand the library of 2D nanomaterials beyond graphene. In this area, an understanding of heterointerface characteristics and mechanisms of charge transfer at 2D materials/neural heterointerfaces (tissue) is vital to help us create dry sensors for flexible neural interfacing. Apart from the material properties, data acquisition at the neural interfaces is also affected by other technical factors. These parameters are not necessarily intrinsic, and they can originate from extrinsic factors similar to the data collection from neighbor neurons that impact the level of noise (SNR). Therefore, it is preferable to understand the logic and important parameters affecting data collection at the neural interfaces. Recently developed dry and flexible ECG neural interfaces based on 2D MXene tattoos yielded higher SNR values compared with those of wet sensors, presenting a bright future for the growth of neural interface technologies based on 2D nanomaterials.

### 4.2. Biocompatibility

Various factors can affect the performance of neural interfaces in a sensing environment. Material characteristics and the fabrication technology can simultaneously affect the performance and durability of 2D materials employed in neural interfaces. The outer layer of skin and epidermis are continuously exposed to a high level of contamination, microbes, dirt, physiological salts and body oils. The biocompatibility issues are more important when an invasive electrode is going to be implemented at the neural interface. In invasive neural interfacing, biocompatibility issues, including inflammatory responses and tissue reactions, are a matter of concern for the fabrication of chronic implantable neural interfaces. An ideal neural interface should minimize the effect of foreign body reactions (FBRs) and perform faultlessly in the human body; therefore, it is necessary to satisfy various requirements during the design of neural interfaces and selection of materials [129]. Due to the direct contact of implanted neural interfaces with body tissue, it is necessary to review the interface reactions that may cause allergenic, toxic or other harmful effects. The intercortical neural interfaces that are in direct contact with body tissue are prone to attack due to the natural reaction of body metabolism. Biocompatibility is an integrated factor that is dependent on chemical composition, mechanical stability, material properties, and surface characteristics. Regarding material/tissue interactions, implanted neural interfaces should not generate external materials or leach chemicals of any form, such as oxidative species and solvents, since these directly affect the recording capability of neural sensors. Several research studies investigated the employment of metals, glass, semiconductors, oxides, hybrid materials, nanostructures and polymers in the development of implantable neural interfaces [129]. The conventional implantable neural interfaces contain metallic components in their structure. The selection of conductor components should be performed carefully. Especially in the case of alloys, it is necessary to prevent the formation of galvanic cell structures to avoid the corrosion of metal components during application in a saline environment. Gold, platinum, tungsten and indium are safe metals with the least inflammatory reactions. Currently, conductive polymers (CPS) with reliable biocompatibility, fast charge transfer characteristics, tunable electrical properties and the capability of drug delivery are being introduced and employed in neural interface technology [129].

The biocompatibility of 2D materials for neural interfacing applications needs to be exploited extensively. So far, most researchers have studied the biocompatibility of graphene neural interfaces. Their studies mostly focused on cell-to-graphene direct interactions. In a typical study, the cellular survival rates on 2D graphene-based substrates were investigated and then compared with other common substrates for neural interface applications [130,131,132,133]. In vitro studies have provided clear insights into the biocompatibility of graphene-based substrates in contact with neural cells. As an example, a 1 month long experiment confirmed that the density of human neural stem cells (hNSCs) on a monolayer CVD graphene (352 ± 20 cell/mm^2^) was tangibly higher than that on a glass substrate (178 ± 27 cell/mm^2^) due to the stronger adhesive characteristics of graphene compared with those of glass substrate [134]. An interesting observation confirmed that the monolayer CVD graphene is more capable of inducing hNSCs to differentiate toward the formation of neurons rather than glial cells [133]. More importantly, it was also confirmed that physiological neural activities, including postsynaptic reactions, short-term synaptic plasticity and synaptogenesis, were not disturbed by 2D graphene substrates [134].

The biostability issues in electronic neurotechnologies are of less concern, since graphene 2D nanosheets are not freely in contact with bio-tissues and are mainly encapsulated by the polymer-based insulating structures. The main group of neural interfaces is composed of multilayer graphene 2D flakes that are randomly oriented into 3D scaffold structures. In this group of neural interfaces, biocompatibility has a vital role in functional applications [135,136,137,138]. Two-dimensional graphene oxide was employed recently as the biofunctional agent in a polyethylene glycol diacrylate hydrogel (PEGDA) scaffold [136]. The GO was used in PEGDA to increase the osteogenic differentiation of adipose-derived stem cells (ADSCs) of the human body. A comparative study between PEDGA and GO-functionalized PEDGA showed that the DNA contents of ADSCs in functionalized samples with GO remained unchanged after 3 weeks of in vitro examination, whereas the DNA content in the non-functionalized PEDGA sample decreased considerably after 2 weeks of examination. Furthermore, the 3 weeks study on RNA expression markers confirmed that the GO functionalized PEGDA samples displayed increased DNA content compared with that in a non-functionalized PEGDA sample. Generally, graphene-based materials and their derivatives are found as chronic implant materials and biocompatible structures for fabrication of neural interfaces based on scaffold structures [53]. However, most of the reported studies are based on animal models; therefore, further experiments are required to reinforce our knowledge on the long-term biocompatibility of graphene-based neural interfaces [53].

Regarding the hydrophilicity properties of 2D materials, recently introduced MXene-based neural interfaces have been found to be capable platforms for the adhesion of neural cells. Recent experiments showed promising biomedical applications for MXene sensors in cancer theranostics [118], diagnostic imaging [139], and biosensing [139,140]. The biocompatibility of MXenes during the immortalization of tumor cells was confirmed during in vitro and in vivo experiments [87,141]. However, only one study [126] has investigated the biocompatibility of MXene film in the neural cell environment. In this study, quantitative measurements of viable neurons grown on Ti_3_C_2_ films showed no significant difference after 7 days in an in vitro experiment. In a comparative measurement between Ti_3_C_2_ and polystyrene substrates, the widespread formation of neural networks on both substrates was confirmed by immunocytochemistry [125]. Overall, the neurons' productivity, adherence and growth on both Ti_3_C_2_ and polystyrene substrates yielded equivalent neuronal viability. These early observations indicate the capacity of Ti_3_C_2_ substrate for the formation and growth of neuronal networks with negligible cytotoxicity effects. However, the mentioned study showed that the neuron cultures on Ti_3_C_2_ experienced a reduction in neuron density compared with that of the polystyrene substrate. This indicates that neural adhesion on the Ti_3_C_2_ substrate decreased during the 7 days experiment. It also suggests that further optimization processes are required to promote neuronal adhesion on the surface of Ti_3_C_2_ as an atypical MXene film [125].

### 4.3. Durability

Another concern is the durability of 2D nanostructures on various substrates. The materials in neural interfaces are prone to delamination, corrosion, breakage, and failure of interconnections. For instance, 2D graphene-based neural interfaces suffer from delamination of 2D graphene nanosheets from the flexible polymer substrates. The Si/SiC substrates were found to be the most stable for long-term performance in bioactive media. However, rigid substrates are not desired for functional neural interfacing applications in the human body. Soft substrates or mesh-like morphologies are possibly the most appropriate options for biosensing and neural interfacing applications. Water-soluble polymer supporting layers (similar to PVA) decorated with functional 2D materials for biosensing are one of the preferable options for the next generation of on-skin dry neural interfaces. Therefore, there is a great opportunity in the fabrication of polymer-based e-skins based on 2D materials for the long-term performance of neural interfaces on human skin. Reusability is another important detail for the desired applications of novel *e*-skins. Specifically, recently fabricated e-textrode neural sensors based on graphitized electro-spun fiber/monolayer graphene clearly demonstrated their stability after washing up to 10 times. Graphene-coated textiles are well-known flexible nanostructures for fabrication of on-skin sensor electrodes for long-term monitoring of EEG signals [86,87,88]. These neural interfaces are mostly developed by the dip-coating of fiber-based textiles similar to nylon, polyesters and cottons in a graphene-containing slurry. These graphene-based coated fibers were used to develop flexible electronic textile or so-called e-textrodes for neural interfacing applications. Just recently, electrospinning techniques enabled the deposition of graphene-based conductive materials on elastomeric fibers to produce flexible wearable electrodes with reliable flexibility and low impedance values [88,142,143]. This technological direction in the evolution of reusable e-skin is still progressing in various areas, from technological concepts to materials in use.

## 5. Conclusions

Generally speaking, the present article provided an overview of recent advancements in the fabrication and development of various types of neural interfaces based on 2D nanomaterials. Both non-invasive and invasive neural interfaces based on 2D materials were introduced, and their functionalities were demonstrated. The multi-functionalities of 2D materials and their physicochemical properties bring both unexpected properties and improved benefits to active nanointerfaces. Recent advances have enabled rapid progress in the fabrication of micron-sized sensors with bifunctional capabilities. Graphene and its family are by far the most employed 2D structures in neural interface technology. Recent advancements in applications of 2D carbon-based nanostructures have enabled the successful monitoring of neural activities and human motion. Apart from graphene-based neural interfaces, researchers have recently focused on other types of 2D materials beyond graphene. Currently, only a few research studies have reported on the application of TMDC and MXene 2D structures in neural interface technology. Therefore, this area represents a great opportunity for the further development and progress of bioelectronic technologies. For successful development in this area of bioelectronics, it will be necessary in the future to address the challenges of fabrication technology, durability and biocompatibility of neural interfaces.

## Figures and Tables

**Figure 1 jfb-14-00035-f001:**
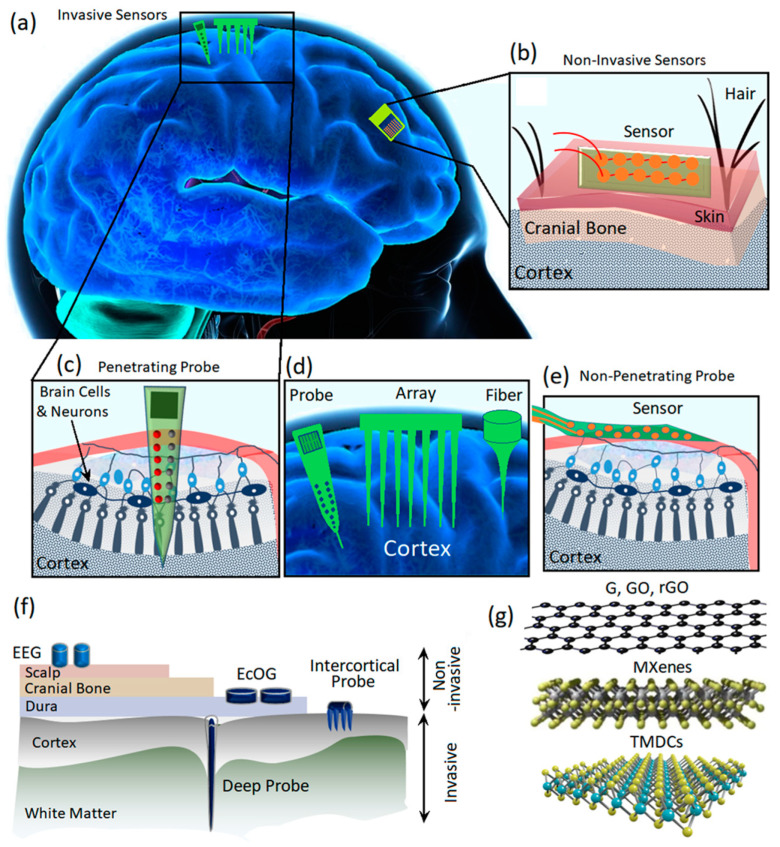
(**a**) The scheme shows invasive and non-invasive neural interfaces assembled on the human brain. (**b**) Non-invasive neural interfaces. (**c**) Implanted (invasive) neural interfaces for investigation of the neural activities of the brain. (**d**) Various types of intracortical probes. (**e**) Non-penetrating invasive neural interfaces. (**f**) Different types of electrodes for neural interfacing and recording of the brain’s electrical activities. (**g**) Various types of 2D materials used as the main components of neural interfaces.

**Figure 2 jfb-14-00035-f002:**
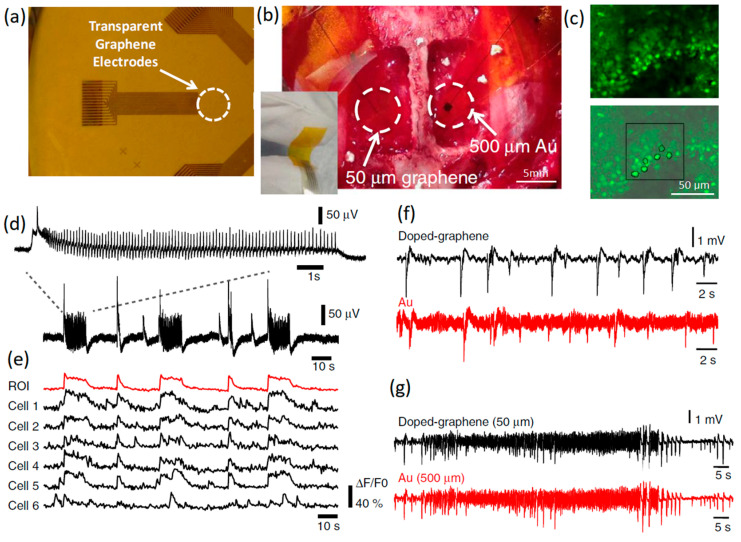
(**a**) Transparent graphene-based invasive multichannel electrode. (**b**) Photograph of flexible graphene electrode on the cortical surface. The scale bar is 5 mm (**c**) Fluorescence calcium imaging from high-frequency neural spikes of hippocampal slice. The scale bar is 50 μm. (**d**) In vitro electrophysiology of hippocampal slice. (**e**) Interictal-like spiking activity recorded by doped graphene. (**f**,**g**) Charge and discharge signals recorded by doped graphene and Au electrodes with similar SNR. Reproduced with permission from Ref. [63].

**Figure 3 jfb-14-00035-f003:**
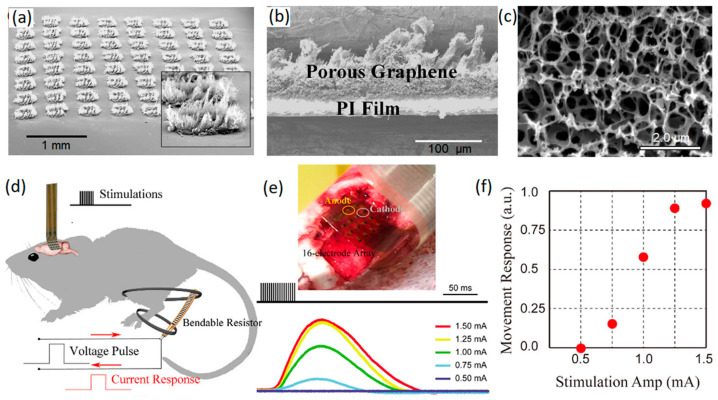
(**a**) Tilt SEM image of porous graphene arrays. (**b**) Cross-sectional SEM image of porous graphene layer. (**c**) SEM morphology of porous surface of electrode. (**d**) Schematic representation of cortical stimulation of ankle and knee flexion. (**e**) Minimally invasive electrode arrays on motor cortex and the recorded stimulus evoking current signals. (**f**) Movement response vs. stimulation amplitude. Reproduced with permission from Ref. [45].

**Figure 4 jfb-14-00035-f004:**
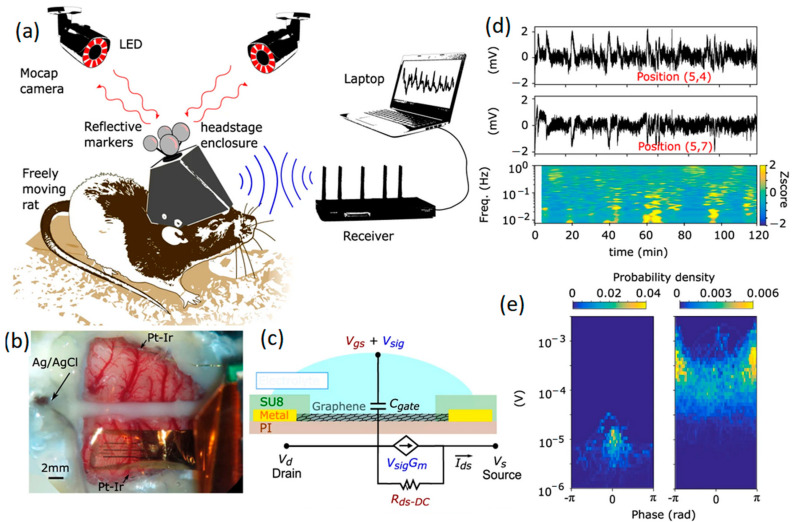
(**a**) Schematic of a rat with implanted untethered recording system. (**b**) GFET multi-arrays positioned on rat cortex. (**c**) Graphical scheme of a GFET with its equivalent circuit. (**d**) Recorded signals extracted from two DC coupled channels. (**e**) Electrode arrays on the motor cortex accompanied by a stimulus evoking current signals. Reproduced with permission from Ref. [74].

**Figure 5 jfb-14-00035-f005:**
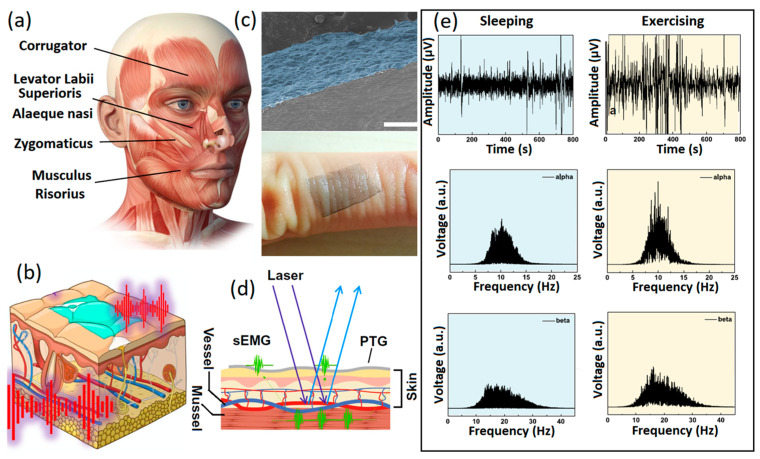
(**a**) Photo depicting the facial expression muscles. (**b**) Graphical image of PTG as skin electrophysiology electrode. (**c**) Cross-sectional SEM image of PTG sensor and its photograph on human skin. (**d**) Simultaneous monitoring of EMG signals and speckle imaging. (**e**) From top down, fast Fourier transformed alpha and beta EEG signals in sleeping and exercising states recorded by PTG skin sensors. Reproduced with permission from Ref. [84].

**Figure 6 jfb-14-00035-f006:**
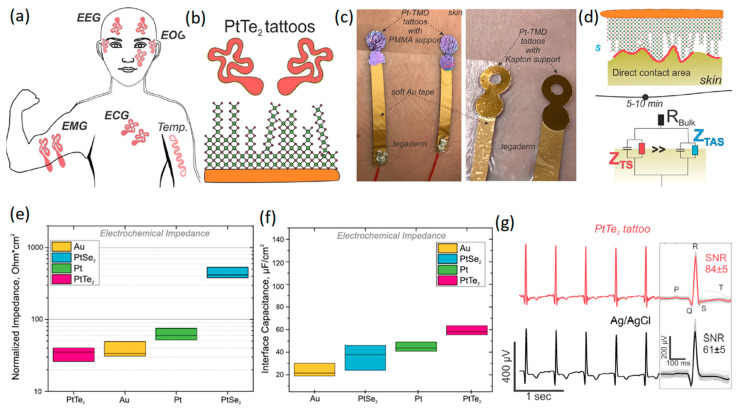
(**a**) Schematic of various types of neural interfaces developed on the basis of 2D Pt-TMDCs. (**b**) PtTe_2_ 2D films grown on flexible substrate. (**c**) Photographs of 2D Pt-TMDCs tattoos, developed on PMMA (left) and Kapton (right) polymeric substrates. (**d**) Schematic representation of the time-related dynamic changes at heterointerfaces between skin and 2D Pt-TMDCs. (**e**) Normalized impedance and (**f**) interface capacitance of 2D Pt-TMDC based neural interface tattoos. (**g**) ECG time trace of 2D PtTe_2_ tattoos and Ag/AgCl gel electrodes accompanied by corresponding SNR values. Reproduced with permission from Ref. [111].

**Figure 7 jfb-14-00035-f007:**
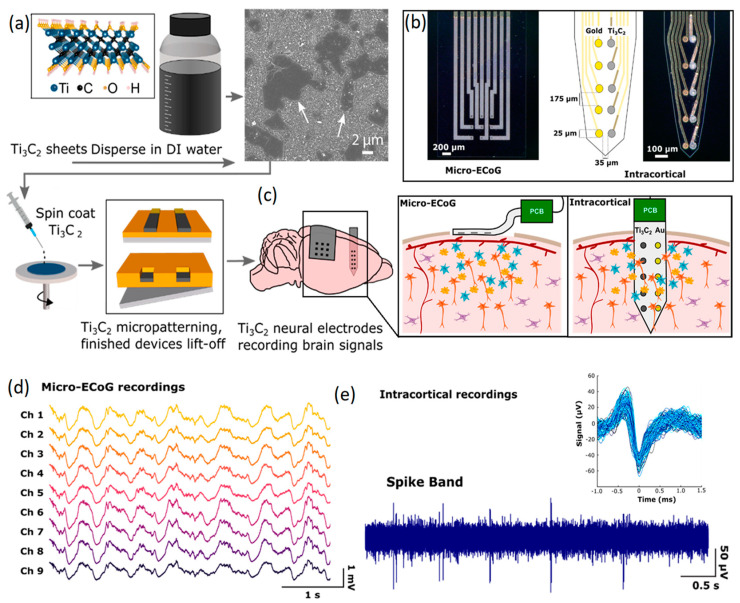
(**a**) Graphical illustration of synthesis process for 2D Ti_3_C_2_ and fabrication of the corresponding neural interface electrodes. (**b**) Images of non-invasive μ-ECoG and invasive intercortical flexible electrodes. (**c**) Schematic representation of in vivo monitoring of neural activities by Ti_3_C_2_ μ-ECoG arrays and Ti_3_C_2_/Au intracortical probe. (**d**) Monitoring of neural activities by Ti_3_C_2_ μ-ECoG and (**e**) Ti_3_C_2_/Au intracortical probe. Reproduced with permission from Ref. [125].

**Table 1 jfb-14-00035-t001:** Neural interfaces based on 2D carbon-based materials.

Neural Interface	Application	SNR	Impedance	Electrode Dimensions	Stability Tests	Ref.
rGO microarrays developed on PDMS (non-invasive)	(e-skin) EEG, EMG, EOG	16.8 dB	~500 kΩ at 50 Hz	N/A	60 times reuse 58 h stability	[20]
Laser-induced GO developed on PET (non-invasive)	ECG	~70 dB	~60 kΩ at 50 HZ	4 cm^2^	>100 h	[21]
Porous graphene array developed by laser pyrolysis (invasive)	Planar electrodes for neural activities	N/A	5 kΩ at 1 KHz	250 μm^2^	10^6^ cycles	[45]
Epitaxial graphene films developed on silicon substrate (non-invasive)	EEG	N/A	68 ± 4 kΩ at 10 Hz	1 cm^2^	120 days	[57]
Doped single-layered graphene (invasive)	Planar electrodes for neural activities	40	541 kΩ at 1 KHz	50 × 50 μm^2^	6 months	[63]
Liquid-crystal GO fiber coated with parylene C	Filament deep probing	N/A	50 kΩ at 1 KHz	N/A	14 days	[77]
CVD graphene-based transparent microarray (invasive)	Optogenetics and neural imaging	N/A	243.5 kΩ at 1 KHz	3.1 × 3.1 mm^2^	70 days implanted	[56]
Multielectrode array based on graphene microtransistor (invasive)	Neural deep probing	N/A	N/A	N/A	10 weeks implanted	[78]
rGO developed on nylon membrane (non-invasive)	ECG, EMG	N/A	~15 kΩ at 1000 Hz	N/A	50 h	[79]
Laser-developed graphene on PU nanomesh (non-invasive)	EEG, ECG, EOG	14.12 dB	N/A	1.5 cm^2^	1000 cycles	[83]
CVD graphene for PMMA tattoo (non-invasive)	ECG, EMG, EEG, EOG	15.22 dB	~13 kΩ at 1 KHz	1.225 cm^2^	N/A	[92]
rGO-PEDOT/PSS fabricated on nylon-lycra (non-invasive)	ECG	21.6 dB	~50 kΩ at 100 Hz	1 × 1 cm^2^	50 cycles	[93]
Electrospun fiber with graphene monolayer (non-invasive)	ECG, EEG, sEMG	~30 dB	150 Ω	N/A	10 times	[94]
CVD graphene multilayer electrodes developed on parylene	Neural interfaces compatible with MRI	N/A	27.4 kΩ at 1 KHz	430 μm diameter	N/A	[95]

**Table 2 jfb-14-00035-t002:** Neural interfaces based on 2D materials beyond graphene.

Neural Interface	Application	SNR	Impedance	Electrode Dimensions	Stability Tests	Ref.
PtSe_2_ and PtTe_2_ based sensors (non-invasive)	EMG, ECG, EOG, EMG	84 ± 6 dB	4.94 ± 1.61 kΩ at 10 kHz	N/A	24 h	[111]
Ti_3_C_2_T_x_ (MXene) based sensors (non-invasive)	EMG signals	~39.23 ± 16.25 dB	~29.78 ± 8.18 kΩ·cm^2^ at 1 kHz	1 cm^2^	N/A	[122]
Ti_3_C_2_ microelectrode array (planar electrodes, non-invasive)	Neural interfacing	N/A	54.6 ± 28.4 kΩ at 1 kHz	7 mm^2^	N/A	[123]
3D mini pillars composed of MXenes (non-invasive)	EEG, EMG, EOG	N/A	2.8 ± 0.9 kΩ at 1 kHz	7 mm^2^	N/A	[124]
Ti_3_C_2_ microelectrode array (intracortical electrodes, invasive)	Neural interfacing	40 dB	219 ± 60 kΩ at 1 kHz	25 μm diameter	7 days in cell culture	[125]
Ti_3_C_2_T_x_ (MXene)	Neural stem cell; Neural spiking, Synaptic transmission	N/A	N/A	0.785 cm^2^	N/A	[126]

## Data Availability

Not applicable.

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
