# Peer review of "Functional Two-Dimensional Materials for Bioelectronic Neural Interfacing"

_jfb, 2023, doi:10.3390/jfb14010035_

Round 1

Author Response

Comments from your reviewer 1:

Authors would like to thank you for your worthwhile comments. They are taken into account in the revised version of manuscript. Please find the detailed responses on the comments. 

  1. There were many grammatical and writing errors in the text that need to be revised and corrected.

Response to comment 1: The English level of manuscript is revised by specialist.

2.1. Figure 1. As mentioned in this sentence “Therefore, the neural interface devices are developed to dynamically interact with the nervous system and brain (Figure 1a) for a wide range of clinical therapy including deep brain stimulation, mind controlling …” does not show anything special. It is better if a part of the brain image is open and shows the interaction of the electrodes with the cells or in some way expresses the issue mentioned in the sentence.

As written in the caption of the figure "Figure 1. (a) The Schemes shows the concept of neural interfacing, and also different type (It is necessary to add an s to the type.) of neural interfaces", does this image really convey this concept or different types of neural interfaces? It seems that it is necessary to make the image more productive and to add expressions related to the different types of interfaces in the brain image.

Response to #2: Figure 1 is totally rearranged and edited. The suggestions are taken into account and the changes are included in new version of Figure 1 and also introduction section. Please see the new version of Figure 1. It contains the concept of brain stimulation and neural interfacing with non-invasive and intracortical sensors.   We further edited the caption of Figure 1a (Types). Two main common types of neural interfaces (Invasive-Intercortical and Non-invasive) are depicted respectively in the Figure 1a, b, c, d, e and f depicting the concept of different type of brain neural interfaces. We also considerably altered the introduction. Please see pages 2, 3, and 4 and new version of Figure 1.

  1. 3. For instance, the external wearable electroencephalogram (EEG), electro-corticogram (EcOG or EOG), and electromyography (EMG) neural interfaces enable…” EOG stands for electrooculogram (a form of EMG conveying gaze direction useful for eye tracking). electrocorticogram (ECoG) is different from electrooculogram (EOG). It is necessary to correct the entire text.

Response to #3:  The authors would like to thank you for mentioning this point. The text is edited now. Please see page 3, line 86-94. It is worth to mention that the introduction is altered considerable.

  1. “However, the sensors used these electrodes suffer from the several fundamental deficiencies including the time-consuming establishment on body organs, skin sensitivity and irritation problems, non-precise performance in moving organs and unstable long-term performance in non-clinical conditions [36-37].”

These fundamental deficiencies are mostly related to the interfaces that are used, for example, in EEG recordings and in contact with the skin, and this problem does not exist in the case of neural interfaces inside the brain. In this field, discussions of corrosion, inflammation, toxicity and biocompatibility become important. It seems that the authors in the whole text should make a distinction between the neural interfaces in the vicinity of the cells inside the body and the interfaces in the skin, and the text should be rewritten based on the difference between these types of interfaces. Or in each part of the text, it should be specified which type of interface it is related to. Due to the many differences between the performance and specifications of these types of interfaces.

Response to #4:  The mentioned point is taken and the changes are included all over the manuscript text.  We specifically amended the manuscript in each part and included the changes regarding the type of interfaces discussing at each section. We also edited the mentioned sentence in the manuscript regarding the challenges of application of Ag/AgCl electrodes. Please see page 4, line 138-144.

  1. “Due to their high surface area, outstanding biocompatibility, easy functionalization and exceptional electronic characteristics, 2D nanostructures are some the main options for development of novel neural interface sensors [3, 56-60].”

Do all 2D nanostructures have high biocompatibility? The claims of this sentence need correction. Considering that the review article is placed as a reference for scientific activities, the sentences and concepts expressed must be carefully selected and written.

Response to #5:  Your valuable point is taken. Not all of the 2D structures represent a high-level of biocompatibility.

Therefore, the manuscript text is altered consequently. Please see page 4 lines 172-178. Furthermore, we include a comprehensive discussions on the biocompatibility issues of the developed neural interfaces based on 2D structures. Please see pages 17, and 18, lines 626-712.

  1. The current review focuses on the latest advancements in the development of neural interfaces based on 2D nanomaterials, including graphene, graphene oxide (GO), reduced graphene oxide (rGO), transition …”

The terms used in the article must be the same. Finally, rGO or RGO that you used in figure 1. d?

Response to #6:  We would like to appreciate you detailed points. We already amended the caption in Figure 1f.  Please see the new version of Figure 1 where rGO is employed as the abbreviation of reduced graphene oxide.

  1. Considering the need for high electrical conductivity in this application, can structures other than rGO, which do not have good conductivity, be used? Or do they end up being converted to rGO as part of the build process?

Response to #7: It is worth to mention that the conductivity level of rGO can also be improved by different techniques including the control of reduction level and functionalization. However, the current review focuses on the last achievements in fabrication of neural interfaces based on 2D materials. There are two research papers that try to find the answer to the above mentioned point.

  • The low-temperature chemical method using hydroiodic acid (HI) and L-ascorbic acid (L-AA) can effectively enhance the conductivity of rGO. As an example, while HI and L-AA are effective reductants separately, the combination of them resulted in rGO films improved electrical conductivity up to 1115 S/m. (please see: 1109/PVSC45281.2020.9300537).
  • In the chemical sensing, the functionalization of rGO with Au nanostructures improved their activity. Please see: Liu T-C, et al. ACS Appl Mater Interfaces 2015, 8:187–196.

As we also discussed in the manuscript, the transition metal dichalcogenides are another replacement for graphene in neural interfaces technology.  We included these points into the manuscript. New references 100 and 101 are included into the manuscript. Please see page 11, lines 414-423 that explain on the above mentioned points.

  1. It is necessary to add some references for the expressions of lines 140 to 170.

   Response to #8:  The new References [61-70] are included in the manuscript to cover the expressions. Please see lines 209-233.   

  1. The letters specifying the different parts of the images are not clear and do not have the right size. On the other hand, there is not the same order in the sequence of these letters and the location of the images, and it creates confusion for the reader in finding the images. It seems that the way the images are arranged needs to be revised.

The details mentioned in the pictures are few and more information needs to be added in the pictures. They are also very small in size, which reduces the resolution of the images.

Response to #9:  The figures are re-arranged and the size of letters are increased. It is worth to mention that we cannot include considerable alteration in the Figure styles since they are also combined from different sources. Please see the new version of Figures and their amended captions.   

  1. The availability of various types of graphene-based materials, including single layer CVD films, multilayered exfoliated films, chemically and mechanically synthesize films and doped graphene nanostructures enabled …”

These terms mentioned in this sentence are different methods of graphene synthesis or different materials?

   Response to #10:  The terms refer to the different methods of synthesis of 2D graphene.  We further include explanations on the sentence to clarify this case.  Please see page 6, line 234-237.

11.” Various conductive layers can be developed on the PDMS layers by electrostatic spinning, sputtering and spray coating to fabricate the skin-mounting side of the neural interfaces. "It is better to use the well-known term electrospinning instead of electrostatic spinning.

Response to #11:  The term is edited to electrospinning. Please see Page 10 lines 374.  

  1. Considering that the area is very influential in the impedance value, in the impedance values reported in the table 1, the value of the electrode area should also be mentioned so that the values are comparable. And considering that you mentioned different applications for electrodes, the frequency value in which you reported the impedance of the electrode should also be mentioned.

Response to #12: This valuable point is taken into account. The new version of Table 1 and Table 2 represent the area and dimension of the electrodes and the frequency of measurements.

  1. What do you mean by the term ECG that you used in parts of the text? Where in the text have you defined this phrase? (As mentioned before, in the entire text and also inside the table, the term EOG should be corrected.)

Response to note #13: ECG is the abbreviated form of electrocardiography or monitoring of heartbeat. Please see corresponding explanations in Page 5, line 201-207. “It is worth to mention that , these 2D based sensors were also employed simultaneously for monitoring of other biological and physical signals of human body, including electrocardiography (ECG or heartbeat), electromyography (EMG, or muscle contraction), and Electrooculography (EOG or eye movement) signals. However, this review specifically cover the last achievements in development of neural interface technologies where 2D materials are employed for monitoring of the brain activities (EEG and EcOG).

Yours faithfully,

Dr. M. K. Akbari

Postdoctoral Fellow

Ghent University Global Campus

Reviewer 2 Report

This review give us the general description of the 2D materials for Neural interface.The development trend in this field is obviously introduced for the readers. All related research aspects contained in literature are described, sorted and analyzed in the paper, upon the interface device R&D parts, e.g. different substrates, 2D interface materials and structures, modification layers, and electrode formation. Specially evaluated intrinsic characteristic for interface are also emphasized such as contact impedance, SNR, interface capacitance, durability,etc. In addition, The introduction of 2D TMDCS thin film structure electrodes research status are also clear and comprehensive. A new promoted development materials for neural interface, MXenes, are also decribled in details for electrophysiology study.

Some words should be considered and corrected, as following:

1.On page 1, Introduction”section, paragrah 1, fifth line,The neural interfaces are the key functions”, the word “are” should be changed to other word such as “execute”;

  On same paragraph, ninth line, what is “prosthetics depression” ?

2.On page 2, nineteenth line, “computer tomography”, maybe "computed tomography".

3.On page 3, second paragraph, fourteenth line, “the warble neural interfacing technology”, it’s better to explain this technology in more detail.

4.On page 4, second paragraph, second line, ”The high temporal resolution”, can be rewrite as "The high temporal resolution recording".

   On same page, third paragraph, fifteenth line, what means “interictal-like”?

5.On page 7, fourth line, “infra-slow”, equally means of "ultra-slow" in the following paragrah context, or not?

   On same page, the second to last line, MΩ μm2 “should be MΩ/μm2?

6.On page 9, Fig.5, the terminology words in the photo (a) could be verified again.

7.On page 12, last paragraph, 17th line, “laser pattering”should be “laser patterning”?

8.On page 15, 3rd paragragh, the fifth to last line, “e-text-rode” , it's better to explain more detail.

Author Response

Dear Charlene Liao

Managing Editor

Special Issue: Selected Reviews in Biomaterials: Development, Applications and Challenges

Journal of Functional Biomaterials

Regarding Manuscript: ID: jfb-209872

Please find the response on comments from review number 2

Comments from your reviewer 2:

General Comment: This review give us the general description of the 2D materials for Neural interface. The development trend in this field is obviously introduced for the readers. All related research aspects contained in literature are described, sorted and analyzed in the paper, upon the interface device R&D parts, e.g. different substrates, 2D interface materials and structures, modification layers, and electrode formation. Specially evaluated intrinsic characteristic for interface are also emphasized such as contact impedance, SNR, interface capacitance, durability, etc. In addition, the introduction of 2D TMDCS thin film structure electrodes research status are also clear and comprehensive. A new promoted development materials for neural interface, MXenes, are also described in details for electrophysiology study.

Response to General Comment: Authors would like to thank you for your promising comments toward our manuscript. Following lines provide the detailed answers on your points.

  1. On page 1, “Introduction “section, paragraph 1, fifth line, “The neural interfaces are the key functions”, the word “are” should be changed to other word such as “execute”;   On same paragraph, ninth line, what is “prosthetics depression”?

Response to #1: The manuscript text is edited. Please see page 1, line 42. The term prosthetics depression is also altered now to more general concept i.e. “neurobiological depression”, please see line 50, and 51 in page 2.

  1. On page 2, nineteenth line, “computer tomography”, maybe "computed tomography".

Response to #2: The term is altered now to “computed tomography”, please see line 127 in page 3.

  1. On page 3, second paragraph, fourteenth line, “the warble neural interfacing technology”, it’s better to explain this technology in more detail.

Response to #3. We presented explanations on the concept of wearable neural interfacing technology. Please see page 5, line 183-194.

  1. On page 4, second paragraph, second line, ”The high temporal resolution”, can be rewrite as "The high temporal resolution recording".  On same page, third paragraph, fifteenth line, what means “interictal-like”?

Response to #4. The manuscript text is altered. The high-temporal resolution is changed to “high-temporal resolution recording”.  Please see page 5, lines 222.

In the case of interictal-like spiking activities please see the following explanations:

Seizure is “a transient occurrence of signs and/or symptoms due to abnormal excessive or synchronous neuronal activity in the brain”. The clinical diagnosis of a seizure is empirical, based upon constellations of certain signs and symptoms, while simultaneously ruling out a list of potential imitators of seizures. Seizures are delimited in time and are labeled as ictal (during a seizure), interictal (between seizures) and postictal (after a seizure) stages.

  1. On page 7, fourth line, “infra-slow”, equally means of "ultra-slow" in the following paragraph context, or not? On same page, the second to last line,”MΩ μm2 “should be MΩ/μm2 ?

Response to note #5. Infra-slow fluctuations (ISFs) or oscillation of EEG signals refers to the specific brain signals with the frequencies within the range of 0.01 Hz to 0.1 Hz. Please see page 8, line 330, 332.

Thank you for mentioning the point on the impedance unit. The unit is also edited in the manuscript. Please see page 9 line 356(MΩ μm-2).

  1. On page 9, Fig.5, the terminology words in the photo (a) could be verified again.

Response to note #6. Authors would like to thank you for mentioning the point. The terminologies are edited now in the new version of Figure 5.

  1. On page 12, last paragraph, 17thline, “laser pattering “should be “laser patterning”?

Response to note #7: We do appreciate your dedication toward the detail of manuscript. The “laser patterning” is edited in the main text. Please see page 15, line 545.

  1. On page 15, 3rdparagraph, the fifth to last line, “e-text-rode”, it's better to explain more detail.

Response to note #8: The corresponding explanations on the e-textrode neural sensors in include in the new version of manuscript. Please see page 19, line 729-740. The new references [154-160] are also included into the manuscript.

Yours faithfully,

Dr. M. K. Akbari

Postdoctoral Fellow

Ghent University Global Campus

Reviewer 3 Report

This paper reviews the recent achievements in 2D-material based bioelectronic systems for monitoring of bio-physiological indicators and bio-signals at neural interfaces. I enjoyed reading the manuscript and recommend it to be published in Journal of Functional Biomaterials. However, prior to acceptance, authors should address the following minor comments:

1. In Figure 1, “(a) The schemes show the concept of neural interfacing, and also different types of neural interfaces.” Where are different types of neural interfaces?

2. In Figure 1d, the atomic structure for graphene is incorrect, and “MXens” should be MXenes.

3. In Introduction, “On contrary to the silicon-based technologies, 2D materials brought essential advancement to the warble neural interfacing technology via their distinguished electronic characteristics, transparency and biocompatibility.” Do you mean “wearable”?

4. In Introduction, “The current review focuses on the latest advancements in the development of neural interfaces based on 2D nanomaterials, including graphene, graphene oxide (GO), reduced graphene oxide (rGO), transition metal dichalcogenides (TMDCs), and MXenes.” There are also other 2D materials, like BP, hBN. Please explain why you chose these 2D materials in this review in Introduction.

5. There are two f in caption of Figure 2.

6. In 2.2, “Consequently, reduced graphene oxide (rGO) with higher conductivity is employed widely for biosensing and especially non-invasive neural interfacing applications.” Will rGO be more hydrophobic and easier to be aggregated in the aqueous? If so, how to balance the advantages and disadvantages?

7. In 2.2, Figure c should be Figure 5c.

8. The format for caption of Figure 6 is not consistent, like (a) (b) and (g).

9. Figure 6 e, f and g were not mentioned in the main text.

10. In 4.2, “MXene based neural interfaces have demonstrated the highest capability for reliable and stable performance in aqueous.” I doubt this statement because MXenes are easy to be oxidized inside aqueous. Please list references to validate this statement.

Author Response

November 29, 2022

Dear Charlene Liao

Managing Editor

Special Issue: Selected Reviews in Biomaterials: Development, Applications and Challenges

Journal of Functional Biomaterials

Regarding Manuscript: ID: jfb-209872

Please find the detailed responses on comments from reviewer number 3.

General Comment: This paper reviews the recent achievements in 2D-material based bioelectronic systems for monitoring of bio-physiological indicators and bio-signals at neural interfaces. I enjoyed reading the manuscript and recommend it to be published in Journal of Functional Biomaterials. However, prior to acceptance, authors should address the following minor comments:

Response to General Comment: Authors would like to appreciate your efforts for improvement of our manuscript.  Following lines provide the detailed answers on your comments.

  1. In Figure 1, “(a) The schemes show the concept of neural interfacing, and also different types of neural interfaces.” Where are different types of neural interfaces?

Response to comment #1: Figure 1 is totally rearranged and edited. The suggestions are taken into account and the changes are included in new version of Figure 1. Please see the new version of Figure 1. It contains the concept of brain stimulation and neural interfacing with intracortical sensors.   We further edited the caption of Figure 1a (Types). Two main common types of neural interfaces (Invasive-Intercortical and Non-invasive) are depicted respectively in the Figure 1a, b, c, d, and e, depicting the concept of different type of neural signal sensing.

  1. In Figure 1d, the atomic structure for graphene is incorrect, and “MXens” should be MXenes.

Response to comment #2: Thank you for your comment. The mentioned points are edited in the new version of Figure 1. 

  1. In Introduction, “On contrary to the silicon-based technologies, 2D materials brought essential advancement to the warble neural interfacing technology via their distinguished electronic characteristics, transparency and biocompatibility.” Do you mean “wearable”?

Response to comment #3: Author would like to thank you for mentioning this point. The text is altered. Furthermore, we added new explanations on the manuscript to clear the concept of wearable neural interfacing technology. Please see page 5, line 183-194.

  1. In Introduction, “The current review focuses on the latest advancements in the development of neural interfaces based on 2D nanomaterials, including graphene, graphene oxide (GO), reduced graphene oxide (rGO), transition metal dichalcogenides (TMDCs), and MXenes.” There are also other 2D materials, like BP, hBN. Please explain why you chose these 2D materials in this review in Introduction.

Response to comment #4: Our manuscript overviews the last findings on the development of neural interfaces based on different two-dimensional materials. Currently, most of the developed neural interfaces are based on graphene 2D structures, and transition metal dichalcogenides. Just recently few scientific evidences mentioned the functionality of MXenes for neural interfacing applications. To the best of our knowledge, there is not any available scientific evidence that reports the application of 2D BP and hBN for neural interfacing. 

  1. There are two f in caption of Figure 2.

Response to comment #5: Thank you for mentioning the point. The caption of Figure 2 is edited now.

  1. In 2.2, “Consequently, reduced graphene oxide (rGO) with higher conductivity is employed widely for biosensing and especially non-invasive neural interfacing applications.” Will rGO be more hydrophobic and easier to be aggregated in the aqueous? If so, how to balance the advantages and disadvantages?

Response to comment #6: the reduction of graphene oxide will retain part of its conductivity, simultaneously it also may loss its hydrophilicity. There is always a balance between this two factors, investigated by other researchers. Nevertheless, please keep in mind that there is no clear evidence about this factor in regard with the performance of rGO in neural interfaces.  It is worth to mention that the conductivity level of rGO can also be improved by different techniques including the control of reduction level and functionalization. However, the current review focuses on the last achievements in fabrication of neural interfaces based on 2D materials. There are two research papers that try to find the answer to the above mentioned point.

  • The low-temperature chemical method using hydroiodic acid (HI) and L-ascorbic acid (L-AA) can effectively enhance the conductivity of rGO. As an example, while HI and L-AA are effective reductants separately, the combination of them resulted in rGO films improved electrical conductivity up to 1115 S/m. (please see: 1109/PVSC45281.2020.9300537).
  • In the chemical sensing, the functionalization of rGO with Au nanostructures improved their activity. Please see: Liu T-C, et al. ACS Appl Mater Interfaces 2015, 8:187–196.

As we also discussed in the manuscript, the transition metal dichalcogenides are another replacement for graphene in neural interfaces technology.  We included these points into the manuscript. New references 100 and 101 are included into the manuscript. Please see page 11, lines 414-423 that explain on the above mentioned points.

  1. In 2.2, Figure c should be Figure 5c.

Response to comment #7. The mentioned part of the text is altered now. Please see Page 11, line 396.

  1. The format for caption of Figure 6 is not consistent, like (a) (b) and (g). Figure 6 e, f and g were not mentioned in the main text.

Response to comment #8. The manuscript is altered. The mentioned captions and corresponding explanations are included into the manuscript. Please see page 13, lines 462-489.

  1. In 4.2, “MXene based neural interfaces have demonstrated the highest capability for reliable and stable performance in aqueous.” I doubt this statement because MXenes are easy to be oxidized inside aqueous. Please list references to validate this statement.

Response to comment #9: Thank you for your technical points. We altered the text and included the practical evidences. The recent experiments showed the promising biomedical applications of MXenes based sensors for cancer theranostics, diagnostic imaging, and biosensing. The biocompatibility of MXenes during the immortalization of tumor cells was confirmed in-vitro and in0vivo experiments. However, there is only one experiment that investigate the biocompatibility of MXenes to neural cells. In this study, the quantification of the number of viable neurons grown Ti3C2 films on cortical neurons showed no significant difference after 7 days of in vitro experiments. In a comparative measurement between Ti3C2 and polystyrene substrates, the immunocytochemistry revealed the widespread formation of neural networks on both substrates. Overall, the neurons productivity, adherence and growth on both Ti3C2 and polystyrene substrates yielded equivalent neuronal viability. These early observations, indicate the capability of Ti3C2 substrata for formation and growth of neuronal networks with negligible cytotoxicity effects. However, the mentioned comparative study showed that the neuron cultures on Ti3C2 experienced a reduction in both neuron density and a commensurate reduction in neurite outgrowth compared to those of on polystyrene substrate. It indicates that the neural adhesion on Ti3C2 substrates decreased during the 7 days of in vitro experiments. It further suggest that the further optimization process are required to promote the neuronal adhesions on the surface of Ti3C2 as atypical MXenes films.

New references are included into the manuscript. Please see page 17, 18, line 626-713. Referents 139-143.  

Yours faithfully,

Dr. M. K. Akbari

Postdoctoral Fellow

Ghent University Global Campus

Reviewer 4 Report

This review is valuable in that it brings to light the possibility of using a wide range of 2-D materials for neural interfacing. I had never really considered MAX phase materials to be candidates for neural interfacing so this is a welcome addition to the literature.

However,

1. The paper desperately requires proof reading. There are so many errors in grammar and spelling, that I stopped marking them.  For the most part the intent of the authors is still clear, but it is most annoying and detracts from the quality of the manuscript

2. the authors should include considerations of spatial resolution. For example in table 1, the impedance measurement is not so meaningful if we dont know the size of the electrode. It does not seem that these 2D materials will be suitable for single neuron recording.  Throughout the manuscript the authors should be careful to specify the size of the electrode because the surface capacitance and impedance, even when quoted per cm2, still can vary a great deal with electrode size. At present it does not seem that these materials will be suitable for single unit recording? 

3. What happens to the materials over time when exposed to the body? Do they induce inflammatory response or gliosis. How does the body react to the MAX phases. Is there any degradation of the surface of the electrodes.

4. In the introductory sections, I think some effort should be devoted to explaining the different measurement modalities. What I mean by this is to explain the physical processes underpinning the measurement. For example, intercortical electrodes are presumably covered in fluid and are not usually in direct contact with neurons but measure voltage spikes via polarization of the interstitial medium. This is presumably quite different from EEG electrodes on the scalp or other electrodes on the skin. Understanding these different modalities may then suggest which 2D material is worth trying out. 

Author Response

December 25, 2022

Dear Charlene Liao

Managing Editor

Special Issue: Selected Reviews in Biomaterials: Development, Applications and Challenges

Journal of Functional Biomaterials

Regarding Manuscript: ID: jfb-2098728

Please find the response on the comments from reviewer number 4.

This review is valuable in that it brings to light the possibility of using a wide range of 2-D materials for neural interfacing. I had never really considered MAX phase materials to be candidates for neural interfacing so this is a welcome addition to the literature. However,

Response to General Comment: Authors would like to appreciate your time and efforts for improvement of the content of our manuscript.  Following lines provide the detailed explanations on your points.

  1. The paper desperately requires proof reading. There are so many errors in grammar and spelling, that I stopped marking them.  For the most part the intent of the authors is still clear, but it is most annoying and detracts from the quality of the manuscript.

Response to comment #1: Authors would like to thank you for the mentioned point. The text is revised to edit the grammar and spelling problems.

  1. The authors should include considerations of spatial resolution. For example in table 1, the impedance measurement is not so meaningful if we don’t know the size of the electrode. It does not seem that these 2D materials will be suitable for single neuron recording.  Throughout the manuscript the authors should be careful to specify the size of the electrode because the surface capacitance and impedance, even when quoted per cm2, still can vary a great deal with electrode size. At present it does not seem that these materials will be suitable for single unit recording? 

Response to comment #2: Your valuable points are taken. The new versions of Table 1 and Table 2 contain the new information, including the Electrode dimensions and frequency of measurements. As you already mentioned, the dimension of electrodes are in the range of cm2 to μm2. Therefore, the comparative results are included into the text. In the case of suitability of the 2D materials for development of single unit recording sensors, there is no practical evidence currently to support this idea.

  1. What happens to the materials over time when exposed to the body? Do they induce inflammatory response or gliosis. How does the body react to the MAX phases? Is there any degradation of the surface of the electrodes?

Response to note #3: We already included the information on the results of stability tests in different environments. Furthermore, we included another section that provides the biocompatibility issues. It is worth to mention that most of the neural interfaces based on 2D materials are based on the graphene 2D families. Therefore, most of the research studies focused on the biocompatibility of graphene based neural interfaces. There are only one case that investigated the biocompatibility of Ti3C2 max phase substrate in neural heterointerfaces. We added the detailed information in Table 1 and 2. Furthermore, we opened up a new parts that reflects the biocompatibility of 2D based materials. Please see page 17, 18, line 626-713. Referents 139-143.

  1. In the introductory sections, I think some effort should be devoted to explaining the different measurement modalities. What I mean by this is to explain the physical processes underpinning the measurement. For example, intercortical electrodes are presumably covered in fluid and are not usually in direct contact with neurons but measure voltage spikes via polarization of the interstitial medium. This is presumably quite different from EEG electrodes on the scalp or other electrodes on the skin. Understanding these different modalities may then suggest which 2D material is worth trying out. 

Response to note #4: Author would like to thank you for this point. The introduction section is totally rewritten. We disclosed different working mechanisms and modalities of neural interfaces in introduction section. Consequently, the Figure 1 is also altered extensively. Please see page 2, 3, 4, and 5.

Please don’t hesitate to contact us, if you have any further questions related to our work.

Yours faithfully,

Dr. M. K. Akbari

Postdoctoral Fellow

Ghent University Global Campus

Round 2

Reviewer 1 Report

All Comments is responded properly and it can be accepted in this stage.